# Activating Telomerase *TERT* Promoter Mutations and Their Application for the Detection of Bladder Cancer

**DOI:** 10.3390/ijms21176034

**Published:** 2020-08-21

**Authors:** Maria Zvereva, Eduard Pisarev, Ismail Hosen, Olga Kisil, Simon Matskeplishvili, Elena Kubareva, David Kamalov, Alexander Tivtikyan, Arnaud Manel, Emmanuel Vian, Armais Kamalov, Thorsten Ecke, Florence Le Calvez-Kelm

**Affiliations:** 1Chair of Chemistry of Natural Compounds, Department of Chemistry, Lomonosov Moscow State University, 119991 Moscow, Russia; 2International Agency for Research on Cancer (IARC), 69372 Lyon, France; lecalvezf@iarc.fr; 3Faculty of Bioengineering and Bioinformatics, Lomonosov Moscow State University, 119234 Moscow, Russia; profard95@gmail.com; 4Department of Biochemistry and Molecular Biology, Faculty of Biological Sciences, University of Dhaka, Dhaka 1000, Bangladesh; ismail.hosen@du.ac.bd; 5Gause Institute of New Antibiotics, 119021 Moscow, Russia; olvv@mail.ru; 6Medical Research and Education Center, Lomonosov Moscow State University, 119992 Moscow, Russia; simonmats@yahoo.com (S.M.); davidffm@mail.ru (D.K.); aleksandertivtikyan@yandex.ru (A.T.); kamalov@mc.msu.ru (A.K.); 7Belozersky Institute of Physico-Chemical Biology, Moscow State University, 119992 Moscow, Russia; kubareva@belozersky.msu.ru; 8Le Creusot Hospital, 71200 Le Creusot, France; manelarnaud69@gmail.com; 9Department of Urology, Protestant Clinic of Lyon, 69300 Lyon, France; emmanuel.vian@gmail.com; 10Department of Urology, HELIOS Hospital Bad Saarow, D-15526 Bad Saarow, Germany; thorsten.ecke@helios-gesundheit.de

**Keywords:** bladder cancer, biomarkers, non-invasive detection, telomerase, somatic mutations, *TERT* promoter region

## Abstract

This review summarizes state-of-the-art knowledge in early-generation and novel urine biomarkers targeting the telomerase pathway for the detection and follow-up of bladder cancer (BC). The limitations of the assays detecting telomerase reactivation are discussed and the potential of transcription-activating mutations in the promoter of the *TERT* gene detected in the urine as promising simple non-invasive BC biomarkers is highlighted. Studies have shown good sensitivity and specificity of the urinary *TERT* promoter mutations in case-control studies and, more recently, in a pilot prospective cohort study, where the marker was detected up to 10 years prior to clinical diagnosis. However, large prospective cohort studies and intervention studies are required to fully validate their robustness and assess their clinical utility. Furthermore, it may be interesting to evaluate whether the clinical performance of urinary *TERT* promoter mutations could increase when combined with other simple urinary biomarkers. Finally, different approaches for assessment of *TERT* promoter mutations in urine samples are presented together with technical challenges, thus highlighting the need of careful technological validation and standardization of laboratory methods prior to translation into clinical practice.

## 1. Introduction

More than 300 thousand new cases of bladder cancer (BC) are diagnosed in the world annually [1]. Currently, the gold standard of BC diagnosis and monitoring is cystoscopy, which is an invasive, painful, and relatively expensive procedure [2]. In the last few years, urine components have attracted the intense focus of investigators aiming to discover novel biomarkers for detection of BC, as urine is in direct contact with potentially malignant urothelium, is easy to obtain, and its testing should potentially be much more cost-effective. 

Molecular markers for such analyses can be of different nature (nucleic acids, proteins, small-molecular-weight compounds), but should be directly linked to the cell processes that are altered during neoplastic transformation (for example, avoidance of programmed cell death through disruption of the telomerase pathway). The characterization of molecular genetic alteration changes, associated with the malignant cell transformation, in differentiation or metastatic potential has provided insights into the interplay of these oncogenic processes and the activation of telomerase and its components [3]. This activation is directly related to the disruption of the cell division control system and, therefore, leads to uncontrolled cell growth. It has been widely demonstrated that telomerase activity is enhanced in 85–90% of tumor cell types [4]. There are multiple underlying mechanisms of telomerase activation [5], but genetic alterations in the promoter region of telomerase reverse transcriptase (*TERT*), leading to both increased gene expression and activity of the enzyme, are considered the most frequent. Therefore, they are considered highly promising putative biomarkers of cancer [6]. The process of reactivating the telomerase enzyme includes alterations in the gene promoter of the catalytic subunit of telomerase caused by methylation [7] and somatic mutations [8], both leading to its overexpression.

In this review, we trace the history of the early-generation of urine BC biomarkers identified in the telomerase pathway, discuss the limitations of the detection of related assays, and highlight the potential of transcription-activating mutations in the promoter of the *TERT* gene detected in the urine as promising simple non-invasive biomarkers for the detection of BC and surveillance for its relapse. Mutations in the *TERT* promoter have been shown to occur in many histological tumor types, making these alterations the most frequent somatic abnormalities detected in cancer so far. In BC particularly, they are detected in 60–85% of cases in all stages and grades of the disease. These mutations have been detected in the intracellular and extracellular DNA fragments from urine samples collected both at the time of primary clinical diagnosis of BC and during post-surgical follow-up. Therefore, they represent promising biomarkers to detect and monitor BC [9]. Recent publications reveal an important short-term clinical perspective of the use of *TERT* promoter mutations as BC urinary biomarkers [10,11]. However, additional validations in large prospective cohort studies and interventional studies are necessary to fully assess their clinical performance and utility. Moreover, it is necessary to evaluate whether the sensitivity of the urinary *TERT* promoter mutations in detecting BC is increased when combined with other urinary biomarkers. Finally, careful technological validation and standardization of laboratory methods for assessing *TERT* promoter mutations in urine samples are critical for their clinical implementation. 

## 2. Telomerase Reactivation in Bladder Cancer and Early-Generation of Telomerase Urinary-Based Biomarkers 

Telomerase reactivation in BC has been first described in the mid-1990s [12]. Telomerase is an RNA-protein machinery that synthesizes repeating telomeric DNA. Telomeres are special structures at the ends of chromosomes where DNA interacts with specific proteins, shaping the “cap” to protect chromosome ends from degradation and to maintain their integrity [13]. Human telomerase contains a protein component (reverse transcriptase or hTERT) and a matrix RNA constituent (telomerase RNA, hTR), which, together with other proteins, form the active holoenzyme whose function is to extend telomeric DNA with new repeats and, therefore, revert the progressive loss of sequences at the ends of chromosomes associated with incomplete DNA replication [14]. In differentiated human cells, telomeres are typically shortened with every cell division up to a programmed critical length that leads to cell aging and apoptosis. During tumorigenesis, the mechanism leading to critical telomere shortening is counteracted by telomerase activation, thus preventing cell death. Enhanced through genetic or epigenetic changes, telomerase activity is, therefore, a telltale sign of malignancy [15]. Bladder cancer cells have shorter telomeres than adjacent normal urothelium [12]. Telomerase activity was shown to be high in BC tumor samples but not in the normal epithelium of patients with BC. Therefore, in many early studies, the measurement of telomerase activity as a biomarker of cancer diagnosis was considered. A relatively simple and accurate test called the Telomere Repeat Amplification Protocol (TRAP) was established to determine telomerase activity in cells and tissue samples [16]. The method relies on the amplification and measurement of the total number of telomeric repeats newly synthesized by the telomerase on a telomere-like oligonucleotide. Due to its direct contact to the urothelium, urine provides the most easily accessible reservoir of potential biomarkers to study urological diseases, so the possibility to use the TRAP assay to detect increased levels of telomerase secreted into the urine by bladder cancer cells has been tested. A case-control study involving 134 primary BC cases and 84 controls demonstrated promising performance of the urine TRAP assay with a sensitivity of 90% and specificity of 88% (when telomerase activity is 50 arbitrary enzymatic units (AEU)) in detecting the presence of bladder tumors in men [17]. While other studies reported a high sensitivity (70–90%) compared to the current standard urine cytology, specificity proved to be lower, ranging from 66% to 88% [17,18,19,20]. The suboptimal of specificity has been attributed to the inherent telomerase activity in inflammatory or non-urothelial cells present in urine samples [21], which results in a significant variability in telomerase activity in urine of healthy individuals but also in cancer patients. This was observed by the same authors who reported an average value of telomerase activity of 27 AEU (total range of 0–88) in urine of healthy individuals and 112 AEU (total range of 30–382) in bladder cancer patients [17]. Another limitation of this method, which possibly reflects the lack of reproducibility between studies, includes the sensitivity of urine samples to inactivating agents that rapidly reduce the activity of the enzyme, giving rise to the need for strict protocols of handling samples at special conditions between sample collection and processing in order to maintain stability of the RNA-protein complex. Therefore, the lack of standardization of the TRAP assay and difficulties due to the technical requirements limit its use as a clinical biomarker for bladder cancer detection [22]. A recent alternative method to determine telomerase activity in human urine samples using the hybridization chain reaction and dynamic light scattering has been developed for the detection of bladder cancer [23]. Preliminary findings indicate a high specificity in cellular models and in few healthy individuals and patients with malignancies other than bladder cancer [24,25,26], but this needs to be confirmed in large case-control studies.

Another existing method to analyze telomerase reactivation is the quantitative measurement of the expression level of telomerase subunits TERT and telomerase RNA (TR) using real-time reverse transcription polymerase chain reaction (qRT-PCR). High expressions of these mRNAs have been observed consistently across many different malignancies, suggesting a promising avenue for early cancer detection in body fluids, especially in urine samples of patients with BC [27]. The quantitative analysis of TR and TERT in urine samples had an overall sensitivity of 77.0% and 55.2%, respectively, and a specificity of 72.1% and 85.0%, and determining both TERT mRNA and TR levels turned out to be more sensitive but less specific than urine cytology [28]. More recently, a protocol of combined modified TRAP and qRT-PCR methods to interrogate urine sediments gave encouraging results for the non-invasive detection of BC [29]. However, obstacles remain before urine telomerase activity-based assays can be translated into clinical practice [30]: (1) A high false-positive rate due to the telomerase activity of blood cells or non-urothelial cells in urine, which, despite attempts to sort out positive non-tumor cells, is yet to be solved; (2) an inconsistent correlation between TERT mRNA and telomerase activity in some tissues [31]; (3) a low number of telomerase-positive cells in urine in early stages of BC, and (4) a possible high rate of telomerase or RNA degradation in the urine and serious technical constraints to maintain their stability. Based on the above considerations, it was of clinical interest to search for novel non-telomerase activity-based urinary biomarkers for the detection and surveillance of BC. Nowadays, several urine-based bladder cancer biomarkers have received FDA-approval: (1) The immunoassays based on the detection of the Nuclear matrix protein 22 (NMP22® BC and its improved variant, the NMP22® BladderChek®, Alere, Waltham, MA, USA) [32] and the detection of the complement factor H-related protein (BTA stat® and BTA TRAK®); (2) the immunofluorescence assays based on the detection of the carcinoembryonic antigen and 2 mucins (ImmunoCyt™/uCyt), and (3) the multitarget fluorescence in situ hybridization (FISH) assay based on the detection of aneuploidy of several chromosomal regions (UroVysion) [33]. Other interesting commercially available biomarkers are emerging: The UBC® rapid (IDL, Bromma, Sweden) test to measure soluble fragments of cytokeratins 8 and 18 in urine [34] and the CxBladder test to identify the presence of five mRNAs (MDK, HOXA13, CDC2, IGFBP5, and CXCR2) in the urine [35]. Only few studies compared the performance of telomerase-based assays with FDA-approved tests for BC detection [32,36,37]. The UroVysion assay (FISH) had higher specificity than the TRAP assays and other urine markers; meanwhile, Bravaccini and colleagues showed that the combination of urine cytology and FISH to the TRAP assay had some potential in discriminating patients with bladder cancer from individuals with other urinary symptoms [36]. Another study conducted in a group of workers employed in the production of tires and, therefore, exposed to various potential bladder carcinogens, and in a control group of unexposed subjects showed that the two-step design using the TRAP assay with standard urine cytology and comet assay as the primary screening tool, and then FISH (UroVysion) in TRAP-positive cases increased the accuracy for the detection of BC as compared to the conventional urine cytology [38].

However, based on performance and cost considerations, none of the commercially available urine biomarkers to date are recommended as reliable diagnostic targets both by the European Association of Urology (link to NMIBC guideline https://uroweb.org/guideline/non-muscle-invasive-bladder-cancer/#5; link to MIBC guideline https://uroweb.org/guideline/bladder-cancer-muscle-invasive-and-metastatic/#6) and American Urological Association (link to NMIBC guideline https://www.auanet.org/guidelines/bladder-cancer-non-muscle-invasive-guideline#x2517) for routine BC clinical management or for screening in high-risk populations [39,40,41,42]. The absence of urine biomarkers that can be clinically exploited and the fact that the re-activation of telomerase is a crucial mechanism of urothelial carcinogenesis (observed in 99% of urothelial carcinomas) rekindled the interest in further research on other markers indirectly influencing telomerase activation, for example, through recurrent genetic changes that have been identified in the regulatory elements of the *TERT* gene. 

## 3. Urinary *TERT* Promoter Mutations: The Holy Grail of a Biomarker for Bladder Cancer Detection and Surveillance? 

### 3.1. TERT Promoter Mutations and Biological Significance in Bladder Carcinogenesis

Since their discovery in 2013 in melanoma samples, mutations in the promoter region of the *TERT* gene have been found to be frequent in several tumor types [43]. Their functional impact has been well-characterized in vitro and associated with the creation of new binding sites to numerous cellular transcription factors, resulting in an increase in *TERT* expression and telomerase reactivation. [44,45]. The introduction of mutations in the *TERT* promoter sequence caused a two- to four-fold increase in promoter activity in reporter cell lines [44]. Thus, detection of such mutations can be seen as an indirect measure of telomerase reactivation and neoplastic transformation of cells.

Two hotspot mutations of the *TERT* promoter have been detected with high frequency in bladder cancer but not in neighboring normal tissues [43,46,47]. These mutations occur at two positions upstream of the transcription starting site, at −124 bp (nucleotide polymorphism G > A, g.1295228 (chr5, 1, 295, 228 assembly GRCh37) or g.1295113 (chr5, 1, 295, 113 assembly GRCh38)) and −146 bp (nucleotide polymorphism G > A, g.1295250 (chr5, 1, 295, 250, assembly GRCh37) or g.1295135 (chr5, 1, 295, 135 assembly GRCh38)) in a GC-rich genome region, which specifies its alternative organization (Figure 1). Reported to be mutually exclusive, somatic mutations in *TERT* promoter occur in 60–80% cases of all stages and grades of BC [40,48,49,50,51,52,53]. Specifically, Kinde et al. were the first to show that *TERT* promoter mutations occur frequently in low-grade, high-grade papillary tumors and carcinoma in situ lesions [54]. Allory et al. reported a *TERT* promoter mutation frequency of 87% in cell lines and of 83% in bladder tumors, regardless of stages or the risk associated with disease. Mutation frequency was virtually the same for low-risk non-muscle-invading bladder cancer (NMIBC) (73%), high-risk NMIBC (74%), and muscle-invading bladder cancer (MIBC) (53%). These mutations occurred more frequently than any other genomic changes in both NMIBC risk categories. *TERT* promoter mutations were not shown to be associated with age, sex, or smoking [48]. In addition to urothelial carcinomas, these mutations have also been reported in other rare histological variants of primary BC, such as squamous cell carcinoma (SCC) [55], small cell carcinoma [56], adenocarcinoma of non-enteric type [57], and plasmacytoid urothelial carcinoma [58].

Furthermore, it has been shown that the two single nucleotide substitutions C228T and C250T together account for 99% of *TERT* promoter mutations in BC. Of the two-thirds of bladder tumors carrying a *TERT* promoter mutation, Rachakonda P.S. and colleagues observed that the C228T mutation (G > A) was the most frequent change in BC followed by the C250T mutation, identified in 53.5% and 11.6% of all tumors, respectively [53]. Two additional rare nucleotide mutations were C228A (number of tumors, *n* = 3) and 57A > C (T > G) −57 (nucleotide polymorphism A > C, g.1295161 (chr5, 1, 295, 161 assembly GRCh37) or g.1295046 (chr5, 1, 295, 046 assembly GRCh38), (*n* = 1)). Mutations in all positions −57, −124, and −146 were mutually exclusive and resulted in the creation of a new common binding site de novo for transcription factors Ets/TCF. Similar results were obtained by Allory and colleagues [48]. The most frequent mutation was C228T (*n* = 65) followed by C250T (*n* = 10), and two additional rare mutations were C242T/C243T (*n* = 2) and C228A (*n* = 1). All mutations were mutually exclusive.

With regard to their potential as prognostic markers, one study investigated the relation between the disease-specific survival of patients with urothelial cancer and (1) the presence of C228T and C250T mutations in the *TERT* promoter; and (2) the level of TERT mRNA expression in two independent cohorts of previously untreated patients (*n* = 35 and *n* = 87). A significant decline in survival was strongly correlated with increased TERT mRNA level, but not with the presence of mutation in the *TERT* promoter [59]. Interestingly, the authors also demonstrated that the presence of a *TERT* mutation was associated with an increase in TERT mRNA expression level, leading to the enhancement in telomerase activity and telomere elongation. It has been hypothesized that this unexpected effect is due to the alternative functions of the telomerase catalytic subunit [60], and the existence of alternative mechanisms, other than mutations in the *TERT* promoter, such as epigenetic changes, also contribute to telomerase activation.

### 3.2. Analytical Methods for Detecting Mutations in the TERT Promoter: Comparison of Analytical Performance and Bias

There is a wide range of established analytical approaches for detecting mutations in the *TERT* promoter. However, the detection of these mutations is complicated by the composition and the structure of the *TERT* promoter genomic region, which is characterized by the highly GC-rich sequence and alternative structures of double stranded DNA in the form of G-quadruplexes, as illustrated in Figure 1 [61]. Another challenge is the detection of low-abundance tumor-derived mutations in body fluids. The DNA fragments carrying the tumor-specific alterations can represent a very small fraction of the total DNA. In blood samples, for example, the circulating tumor DNA fraction has been reported to be as low as 0.5% [62]. The analytical sensitivity of the assays is, therefore, critical in such settings. Despite such constraints, many quantitative PCR-based diagnostics described below have been successfully applied to human biological fluids, e.g., whole blood [63], urine (see Table 1), and urine samples of patients with hematuria (Appendix A).

Currently, the used method for detecting known genetic changes in clinical research is real-time PCR (quantitative PCR and its modifications). Yet, they often lack the sensitivity to detect underrepresented genetic alterations, so-called low mutant allelic fraction (MAF), which are often found diluted with wild-type DNA fragments originating from non-malignant cells or non-mutated cells. Furthermore, the heterogeneous composition of DNA fragments found in ‘liquid biopsy’ samples may also complicate the analysis. Studies that used real-time PCR to detect *TERT* promoter mutations in tumor and urine samples are shown in Table 1. Successful application of a qPCR-based method known as castPCR was described by Wang and co-workers [61]. In comparison to Sanger-sequencing, castPCR demonstrated dramatically higher sensitivity and specificity in a wide range of tumors of the urinary system (Table 1). The most recent progress with regard to the detection of *TERT* promoter mutations by qPCR has been achieved by Batista and colleagues who demonstrated that their sensitive, urine-based assay called Uromonitor^®^ (Uromonitor Maia, Portugal) based on competitive allele-specific discrimination PCR was capable of detecting trace amounts of *TERT* promoter mutations in urine samples [65].

Next-generation sequencing (NGS) can simultaneously analyze millions of DNA copies. The identification of low-allelic somatic mutations requires ultra-deep sequencing so that the few sequencing reads with the mutant allele can be generated within the pool of wild-type reads. To achieve such high sequencing coverage of the screened genomic region(s), sequencing must be targeted. Traditional NGS-based targeted sequencing is able to detect mutant DNA forms at or higher than 2% allelic fraction against the background of the wild type DNA [66], but recent developments of NGS systems, such as Safe-SeqS [54,67], Tam-Seq [68,69], and CAPP-seq [70], improved threshold limits, the latter reaching an analytical sensitivity of 0.0025% MAF. Avogbe et al. recently developed UroMuTERT, a simple, non-invasive, and sensitive NGS-based assay for the detection of low-level *TERT* promoter mutations. Combined with a specific algorithm developed by the same group, called Needlestack [71], UroMuTERT achieved detection thresholds of 0.8% and 0.5% mutant allelic fraction MAFs for C228T and C250T mutations, respectively [10].

In addition to the next-generation sequencing methods, another platform that can detect low-abundance mutant DNA molecules against a background of the thousand-fold excess of wild-type molecules is droplet-digital PCR (ddPCR). It combines the short hands-on-time and easy laboratory workflows and does not require complex bioinformatic analysis (Table 1), making it highly suitable for implementation into clinical practice.

Figure 1 shows a possible organization of the *TERT* promoter region. This model was generated in the online server “QGRS Mapper” and the particular fragment shown on the picture was described [61] to determine mutations C228T and C250T in upper tract urothelial carcinomas (UTUC) using PCR in conjunction with subsequent Sanger sequencing. All the analytical approaches described above include an amplification step of genomic regions containing −124 and −146 sites from the ATG starting codon, whose length may vary according to primer design. The occurrence of C228T and C250T mutations can distort the double-stranded structure of this region, and the amplification efficiency may also be subject to the ability of primers to anneal and extend template DNA in such complex regions with secondary structures. This could explain the wide ranges of reported sensitivities and specificities. 

To develop diagnostic approaches based on the identification of mutations in the promoter region of the *TERT* gene, it will be critical to compare the performance of screening methods and provide harmonized and standardized laboratory procedures. 

### 3.3. Predictive Significance of Determining TERT Promoter Mutations in Urine

The high frequency and the localization of mutations in a small region of the *TERT* promoter provided an extraordinary opportunity for a simple non-invasive assay for early detection or monitoring the recurrence or progression of disease in the patients whose tumors carry one of those variants. This is especially pertinent for the conception of an early detection test as *TERT* promoter mutations have been reported to be early events in the BC tumorigenesis process [54]. Normal urothelium cells and extracellular DNA (also called cell-free DNA or cfDNA) are constantly released into the urine. Malignant transformation of bladder tissue will lead to exfoliated tumor cells and circulating tumor DNA (also called ctDNA) to mix with normal cells and cfDNA in the urine. 

*TERT* promoter mutations (C228T and C250T) have been previously detected in DNA from urinary exfoliated cells (cellDNA) collected prior to diagnosis and during post-surgical follow-up, with sensitivities and specificities varying from 52% to 82% and from 83% to 99%, respectively, in patients with incident or early BC and from 42% to 74% and 73% to 93%, respectively, in patients with recurrent BC [48,54,72,73,75,77]. Two studies reported a sensitivity of 80% using pre-surgery urine cellDNA but no information was provided on the primary or recurrence status [46,50]. The first indication these mutations detected in urine samples during follow-up was associated with recurrence and, therefore, could potentially serve as markers to monitor the disease status that was provided by the study conducted by Kinde et al. [54]. While limited in size, the authors showed that among patients whose tumors harbored *TERT* promoter mutations (*n* = 11), the same mutations were present in urine collected for follow-up in seven of eight patients with relapse but in none of the six patients without recurrence [54]. In line with these findings, Descotes and colleagues showed that, in particular, the presence of *TERT* mutant DNA forms in post-surgical urine samples was associated with recurrence in 100 patients initially diagnosed with NMIBC [46].

The association held true in a limited subset of patients with negative cystoscopy (*n* = 6), suggesting that *TERT* promoter mutations in urine could be a promising avenue for early detection of recurrence in patients under surveillance for BC. The same research group also showed that the detection of urinary mutations could be used as a dynamic monitoring of recurrence. This was illustrated in one patient for whom the absence of the initially detected C250T mutation was noted in post-diagnostic serial urine samples for 7 years before being detectable at the time of recurrence confirmed by cystoscopy. However, Allory et al. reported a relatively low specificity for the prediction of recurrence as mutations were detected in 27% of recurrence-free patients under surveillance for BC [48]. The high false-positive rate may reflect a timeline that is suboptimal for follow-up, which, if prolonged, may contribute to increase specificity, as patients with clinically undetectable tumors at the time of a *TERT* promoter mutation positive test may present with clinically detectable tumors later on. More well-powered longitudinal studies with sufficient follow-up durations and serial post-surgery urine samples are required to fully assess the true performance of these biomarkers for the prediction of BC recurrence. 

There is growing evidence supporting the utility of urinary *TERT* promoter mutations to detect primary BC. Allory and colleagues first reported a sensitivity of 62% for the detection of primary BC with a specificity of 90% in individuals with hematuria but no bladder tumor [48] (Table 1 and Appendix A). Combining urinary *TERT* promoter mutations with other DNA-based markers was also evaluated [72,75]. In a prospective blinded study, urine samples from 475 patients with gross hematuria collected at the time of standard urological examination (flexible cystoscopy and computed tomography urography) were tested for DNA mutation (*TERT* and *FGFR3*) and methylation biomarkers (*SALL3*, *ONECUT2*, *CCNA1, BCL2, EOMES,* and *VIM*) to determine whether a urine-based DNA test could replace flexible cystoscopy in the initial assessment of the most common BC symptom, i.e., gross hematuria. Of the 99 (20.8%) patients presenting urothelial bladder tumors, the DNA test had a sensitivity of 97.0% and a specificity of 76.9%. Detection of mutations in the *TERT* promoter showed the highest sensitivity (81.8%), but at the same time, the lowest specificity (83.5%) for individuals with hematuria [72] (Table 1 and Appendix A). The *FGFR3* gene is the most frequently mutated gene in NMIBC with a total frequency of 70%. While they are much less frequent than *TERT* promoter mutations, they could still represent a putative interesting combined biomarker for the detection of BC. The added value of their combination with *TERT* promoter mutations for the comprehensive non-invasive detection of BC has to still be demonstrated in independent study. A combined DNA-based biomarker approach was also recently evaluated in a screening study conducted by Springer et al. where they assessed the performance of a multigene panel assay that includes the screening of *TERT* promoter mutations and regions of interest in ten other somatically mutated genes (UROSEEK) for detecting BC 0–18 months prior to clinical diagnosis in high-risk symptomatic patients. The authors reported a sensitivity of 83% and a specificity of 93% for their panel, while *TERT* promoter mutations were detected in 57% of the cases. Specifically, the sensitivity and specificity of the *TERT* promoter mutation in urine samples of individuals with hematuria were 55% and 90%, respectively (Appendix A) [75]. In a recent case-control study, Avogbe et al. used their developed single-plex ultrasensitive UroMuTERT assay to test the urinary DNA samples (both cfDNA or cellDNA) of 93 primary and recurrent cases with urothelial cancer and 94 controls, and compare its performance to that of urine cytology for the detection of urothelial cancer [10]. C228T or C250T mutations were detected in urinary cfDNA or cellDNA with 87.1% sensitivity and 94.7% specificity. The UroMuTERT sensitivity was consistent across primary and recurrent cases, and tumor stages and grades, and highest for urinary cfDNA and cellDNA combined. It also significantly outperformed the sensitivity of urine cytology, especially for detection of low-grade early-stage urothelial cancer [10]. In addition, the UroMuTERT single-gene assay demonstrated comparable performance to that of the UroSEEK multiple markers assay (including C228T and C250T) for the detection of primary or early urothelial cancer (sensitivity of 86.7% versus 83%; specificity of 94.7% versus 93%). Therefore, more studies are required to understand whether the observed differences in the detection rate of urinary *TERT* promoter mutations may originate from pre-analytical procedures or from the use of multiple urinary DNA sources versus one or from differences in prevalence of *TERT* promoter mutations in BC across populations. This has important implications as a simple single-gene assay with harmonized and standardized procedures for urine collection and processing might be able to achieve the same clinical performance for the detection of BC as complex multi-gene assays, which are more expensive and clinically less easily implemented. 

While most studies conducted so far have focused on the evaluation of urinary cellDNA, there is, in addition to what Avogbe and colleagues reported [10], accumulating evidence that urinary cfDNA could be a reliable alternative source of urinary DNA for non-invasive genomic profiling of BC. While being based on recurrent clinically actionable genomic aberrations rather than on the assessment on *TERT* promoter mutations, a study reported that the use of urinary cfDNA led to higher analytical sensitivity (90%), as well as the use of urinary cellDNA (61%) for the detection of UC tumor-associated alterations [78]. These findings are in line with an initial study from 2007 reporting the superiority of urinary cfDNA over cellDNA for the detection of genetic alterations of patients with urothelial cancer [78,79]. Applied to the detection of the *TERT* promoter mutations, two recent studies highlighted the potential of the marker in urinary cfDNA for the detection of BC [80,81]. Specifically, of 77 patients whose tumor cells carried the 228 G > A/T mutation, the same mutation was detected by ddPCR in urinary cfDNA of 71 individuals (92%) and the mutation was absent in cfDNA of 26 of 27 healthy patients (specificity of 96%). Patients with false-negative results had an early-stage tumor, and increased mutant allelic fraction was found to correlate with increased stages of the disease. Concordant mutational status between tumor tissues and liquid biopsy was obtained in 92% of cases [80]. However, in studies comparing the analytical sensitivity of both forms of urinary DNAs, the results are sparse. Stasik and colleagues demonstrated a better sensitivity in using cellDNA (83%) than in using cfDNA (77%), but, overall, the *TERT* mutation allelic frequencies (MAF) were highly correlated, suggesting little added value in using cfDNA as an alternative source of urinary DNA [2]. This observation is in agreement with the results from Ward et al. who demonstrated an equal ability to detect somatic tumor mutations in cfDNA and cellDNA [82]. Avogbe and co-workers also reported an overall high concordance between cfDNA and cellDNA results but still observed the highest sensitivity for the combined source of DNA (87.1%) as opposed to cfDNA only (81.8%) and cellDNA (83.5%), highlighting the potential utility of combining multiple sources of DNA for the assessment of the marker in rare cases presenting with discordant results between cfDNA and cellDNA [10]. Interestingly, Stastik et al. observed a potential advantage of using urinary cfDNA in leukocyte-rich urines where the mutant allelic fractions of *TERT* promoter mutations were higher in cfDNA than in cellDNA [2]. 

In order to validate promising biomarkers, expert groups recommend a nested case-control study design within prospective cohorts in which samples collected at enrolment within the targeted population will be tested for the biomarker(s) in asymptomatic individuals who will develop cancer later and those who will not [83,84]. This sort of study was recently conducted by Hosen et al. who investigated the potential of urinary *TERT* promoter mutations as early detection biomarkers for bladder cancer in asymptomatic individuals in a case-control study nested within a longitudinal population-based prospective cohort of 50,045 Iranian individuals (the Golestan Cohort Study). *TERT* promoter mutations were assessed in baseline urine samples (1.9–4.5 mL) from 38 individuals who subsequently developed primary BC and 152 matched controls using the UroMuTERT and droplet digital PCR assays. Sequencing results were obtained for 30 cases and 101 controls. *TERT* promoter mutations were detected in 14 pre-clinical cases (sensitivity 46.67%) and none of the controls (specificity 100.00%). Most notably, the mutations were detectable up to 10 years prior to clinical diagnosis, indicating that detecting pre-clinical BC using cost-effective urinary *TERT* biomarkers may provide a valuable opportunity for BC screening and management [11].

Avogbe and colleagues [10] developed a predictive assay UroMuTERT, based on NGS (single-plex assay) of the hTERT promoter and the certain algorithm for detecting mutations of low-allelic fractions. Mutations in the *TERT* promoter in the urine DNA (cfDNA or cellDNA) showed superior sensitivity and specificity compared to all the methods described above, significantly surpassing the urine cytology especially for detecting early-stage NMIBC, which allowed the authors to propose modifications to the classic diagnostic protocol. The high recurrence rates of bladder cancer require frequent follow-ups involving expensive and invasive cystoscopic examination, thus further increasing the already high initial expenses for the management of bladder cancer [85,86]. The average costs of cystoscopy are around $206, and the cost of non-invasive urine cytology is around $56 [87]. By contrast, the cost of the NGS-based and ddPCR assays for detecting urinary *TERT* promoter mutations in bladder cancer developed by our group [10,11] is about 24€ per sample, and, therefore, has the potential to be easily implemented for cost-effective bladder cancer management strategies. 

## 4. Conclusions

In summary, urinary *TERT* promoter mutations have demonstrated significant potential to be used as reliable, inexpensive, and non-invasive biomarkers for early detection and monitoring of BC. Moreover, urine ddPCR-based assays have been shown to be capable of detecting very low levels of these mutations, cost-effective, and simple to use, and would therefore represent an attractive method for clinical practice. The fact that *TERT* promoter mutations have been identified in urine years prior to the primary clinical diagnosis of BC and in some relapse-free patients under surveillance reflects the early occurrence of the mutations in the primary carcinogenic and in the relapse processes, providing a window of opportunity for early molecular detection and intervention. It may also explain the lack of specificity in some studies with the insufficient duration of follow-up. Therefore, large studies with a long-duration follow-up should further assess the robustness of these biomarkers for both detection and surveillance of BC. In particular, it should be evaluated whether a clinical diagnosis can be made through cystoscopy or urography in asymptomatic individuals or patients under surveillance presenting with a positive urinary *TERT* promoter mutations assay, or they would benefit from regular *TERT* mutation screening until the tumor becomes detectable.

Studies have shown that screening the high-risk population for bladder cancer with robust urinary markers, while not recommended by urological societies at present, could be cost-effective. Should their clinical relevance be demonstrated in individuals at high-risk of developing the disease (i.e., subjects with symptoms, mainly hematuria and/or lower urinary tract symptoms, or subjects with occupational exposure to certain chemicals), this may increase awareness of bladder cancer risk and facilitate the implementation of screening strategies in defined high-risk groups who would benefit from close surveillance with a non-invasive test. Furthermore, early detection of primary or recurrent BC using urinary *TERT* promoter mutations as a primary tool should lead to timely therapeutic intervention and better survival. It should also reduce both the numbers of unnecessary cystoscopy procedures in patients with a *TERT* promoter mutation negative test and the cost of clinical management of suspected BCs. In addition, as it is unlikely that *TERT* promoter mutations in BC could be detected in urine in all BC cases, it would be important to evaluate whether the clinical performance of this promising and already successfully applied biomarker could further increase when combined with existing urinary biomarkers, which alone lacks the sensitivity and specificity for clinical utility.

Finally, the origin of the occurrence of the *TERT* promoter mutations and their correlation to the BC phenotype still have to be elucidated. Future research on the etiologies of mutations occurrence in certain parts of the genome leading to enhanced activity of the *TERT* promoter will result in a new practical understanding of the biology of BC and possibly the development of preventive approaches. With regard to potential therapeutic applications, it is worth noting that the region of the *TERT* promoter that frequently carries *TERT* promoter mutations in BC, which are absent in normal bladder cells, can presumably become the target of anti-cancer therapy, including novel TERT-based immunotherapies, which could be tailored to patients whose tumors harbor these mutations [88,89].

## Figures and Tables

**Figure 1 ijms-21-06034-f001:**
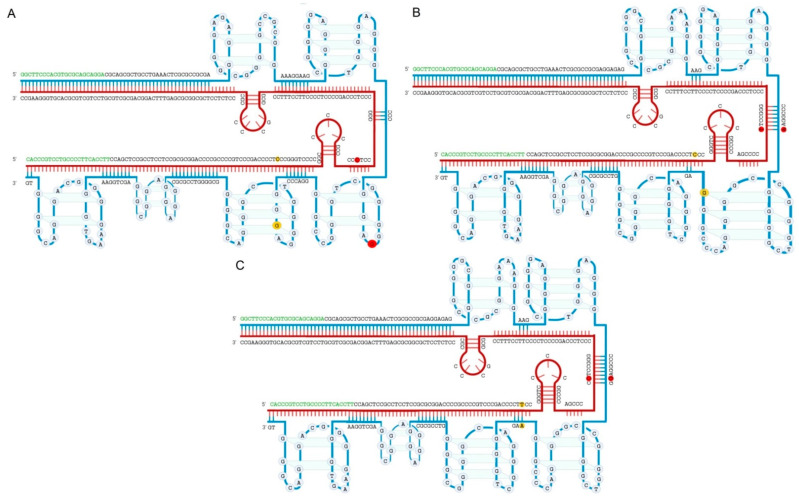
Predicted intramolecular distribution of putative Quadruplex forming G-Rich Sequences (QGRS) in the *TERT* promoter sequence. (**A**) the structure of wild type (WT) DNA; (**B**) only with mutation C228T; (**C**) only with mutation C250T. The two nucleotides at −124 and −146 positions from the ATG start site are highlighted with red and yellow circles respectively. These two hotspots affected by mutations predominantly including C228T and C250T are of the interest for both fundamental research and clinical diagnostics. G-strand and C-strand are marked in blue and claret lines respectively. Regions corresponding to the primers applied to amplify this fragment are marked in green. Despite the close resemblance between all three types of structures they differ substantially. The structures were obtained using online programs “QGRS Mapper” [64] (output data are provided in Appendix A). In this manner both G- and C-strands were analyzed, however only G-strand contains G-quadruplexes. Procedure and software used to calculate and create the structures are described in the Appendix A in more detail.

**Table 1 ijms-21-06034-t001:** Accuracy and methodological characteristics of tests for detecting the *TERT* promoter mutations in the urine for various neoplasias of the urinary system.

Article	Tumor Type	Method	Number of Patients	Size of Control Group	Sensitivity %	Specificity %	Length of PCR Product	Primers (Sequences Are Presented from 5′ End to End) and Probes
[56]	Small cell carcinoma (SCC)	PCR+ Sanger sequencing	11	3	100	100	163	CAGCGCTGCCTGAAACTC; GTCCTGCCCCTTCACCTT
[61]	Ureter carcinoma (UC)	PCR+ Sanger sequencing	20	0	94	10	193	CACCCGTCCTGCCCCTTCACCTT; GGCTTCCCACGTGCGCAGCAGGA-
	Renal pelvic carcinoma (RPC)		16	0	93.8	25	193	
	UTUC (RPC + UC) C228T	PCR+ Sanger sequencing	10	37	60	97	193	
		castPCR	10	37	90	92	dnp *	dnp
	BC (C228T)	PCR+ Sanger sequencing	36	33	47	100	193	CACCCGTCCTGCCCCTTCACCTT; GGCTTCCCACGTGCGCAGCAGGA
		castPCR	36	33	86	97	dnp	dnp
	UTUC + BC	PCR+ Sanger sequencing	46	70	50	98	193	CACCCGTCCTGCCCCTTCACCTT; GGCTTCCCACGTGCGCAGCAGGA
		castPCR	46	70	89	96	dnp	dnp
[10]	Urothelial cancer (UC) primary	UroMuTERT (NGS)	45	94	86.7	94.7	147	CTTCCAGCTCCGCCTCCTCCGCGCGG; AGCGCTGCCTGAAACTCGCGCC
	Urothelial cancer (UC) recurrence		48	94	87.5	94.7	147	
	UC (Diaguro)		93	94	87.1	94.7	147	
[72]	Urothelial bladder carcinoma	ddPCR	99	376	81.8	83.5	52	C228T: CGGAAAGGAAGGGGAGGG;GTCCCCGGCCCAGC
								Mut: [6FAM]-CCC+C+T+T+CCGG-[BHQ_1]
								WT: [HEX]-CCCC+T+C+CGGG-[BHQ_1]
							60	C250T: TGGGAGGGCCCGGAG;GACCCCGCCCCGT
								Mut: [6FAM]CCC+C+T+T+CCGG[BHQ_1]
								WT: [HEX]CCCC+T+C+CCGG[BHQ_1]
[55]	Squamous cell carcinoma Benign transurethral bladder biopsy samples	Safe-SeqS	15 0 ^i^	94 ^ii^ 8	80	dnp	125 dnp	1st couple: CACACAGGAAACAGCTATGACCATGGGCCGCGGAAAGGAAG; CGACGTAAAACGACGGCCAGTNNNNNNNNNNNNNNCGTCCTGCCCCTTCACC ** 2nd couple: CACACAGGAAACAGCTATGACCATGGCGGAAAGGAAAGGGAG; CGACGTAAAACGACGGCCAGTNNNNNNNNNNNNNNCCGTCCCGACCCCTC
[73]	NMIBC primary	SNaPshot assay	230	0	69	52	dnp	dnp
[48]	BC (primary)	SNaPshot assay	118	0	62	–	155	AGCGCTGCCTGAAACTCG; CCCTTCACCTTCCAGCTC
	BC (recurrence)		113	0	42	–	155	Probes: for C228T/A T_23_GGCTGGGAGGGCCCGGA
	BC (recurrence-free samples)		0	218	–	73	155	for C250T T_39_CTGGGCCGGGGACCCGG
[74]	Renal pelvic carcinoma (RPC)		5	0	60	dnp	193	CACCCGTCCTGCCCCTTCACCTT; GGCTTCCCACGTGCGCAGCAGGA
	UTUC		14	0	29	dnp	193	
	Chromophobe renal cell carcinoma (CRCC)		8	0	13	dnp	193	
	Ureter carcinoma (UC)		9	0	11	dnp	193	
	Clear cell renal cell carcinoma (CCRCC)		96	0	9.3	dnp	193	
	Renal cell carcinoma ^iii^ (RCC)		109	0	9.2	dnp	193	
[75]	BC early detection	PCR + Illumina sequencing	570	188	57	99.4	126	GGCCGCGGAAAGGAAG; CGTCCTGCCCCTTCACC
	UTUC		56	188	29	99.4		
	BC surveillance		322	188	57	99.4		
[11]	BC	UroMuTERT and ddPCR	30	101	46.7	100	65	C228T: CCCTCCCGGGTCC; CCGCGGAAAGGAAGG; probes: Mut: CCCGGAaGGGGCTG (FAM_lowaBlack); WT: CGGAgGGGGCTGG (HEX_IowaBlack). C250T CTTCACCTTCCAGCTCC; GAGGGCCCGGAGG; probes: Mut: CCCGGaAGGGGTCG (FAM_lowBlack); WT: ACCCGGgAGGGGT (HEX_IowaBlack).
[47]	UTUC	ddPCR	56	50	46.4	96	113	dnp
[76]	BC (supernatant) BC (sediment) Non-cancer hematuria	NGS	92	0	46	100		NGS-primers: ACCTTCCAGCTCCGCCTCCTCCGCGCGGAC; AGAGGGCGGGGCCGCGGAAAGGAAGGGGAG
			92	0	48	100		
			0	33				
[57]	Primary bladder adenocarcinoma	Safe-SeqS	14	94 ^iv^	28.6	dnp	125	1st couple: CACACAGGAAACAGCTATGACCATGGGCCGCGGAAAGGAAG; CGACGTAAAACGACGGCCAGTNNNNNNNNNNNNNNCGTCCTGCCCCTTCACC 2nd couple: CACACAGGAAACAGCTATGACCATGGCGGAAAGGAAAGGGAG; CGACGTAAAACGACGGCCAGTNNNNNNNNNNNNNNCCGTCCCGACCCCTC
	Benign transurethral bladder biopsy samples		0	8			dnp	
[53]	Urothelial cell carcinoma	PCR+ Sanger sequencing	327	0	dnp	65.4	343	AGCACCTCGCGGTAGTGG; GGATTCGCGGGCACAGAC

* “dnp”–data not provided; ** “N” is a degenerate base (it can be A, T, G, or C with equal likelihood); ^i^. Table fields with “Number of patients 0” correspond to control group or additional control group; ^ii^. Peripheral blood; ^iii^. In this study, 6 subtypes of RCC tumor were investigated. In the table data only about 2 subtypes (ccRCC and chRCC) are presented. 4 remaining RCC tumors did not harbor *TERT* promoter mutations; ^iv^. Peripheral blood.

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
