# Peer review of "Activating Telomerase TERT Promoter Mutations and Their Application for the Detection of Bladder Cancer"

_ijms, 2020, doi:10.3390/ijms21176034_

Round 1
Reviewer 1 Report
This review describes details on detecting performance of TERT-based assays. This is a nice review providing comprehensive information on the topic.
The authors may want to address the following comments.
1) The authors may want to provide a table summarizing performance of TERT-based assays to detect bladder cancer among patients with hematuria.
2) Conclusion: The authors mention that TERT-based assay is cost-effective. However, they did not describe cost of the assay. Please provide the basis of the conclusion.
Author Response
“This review describes details on detecting performance of TERT-based assays. This is a nice review providing comprehensive information on the topic. The authors may want to address the following comments.”
Point 1: The authors may want to provide a table summarizing performance of TERT-based assays to detect bladder cancer among patients with hematuria.
Response 1: All described assays are based on PCR and its modes. Moreover, all these base methods are applicable both for urine samples and for blood samples including whole blood [1,2]. DNA samples obtained from cell lines or patients’ tissues are exposed to preliminary purification imperatively. After DNA purification none of potentially interfering substances remains in the analyzed samples. Undoubtably all mention test-systems are appropriate for cancer diagnostics even in patient with hematuria. Indeed, not all papers contain explicit indication of the feasibility to use these assays for patients with hematuria but it is implicit. The prepared table is adduced into the supplementary file “Table S4. Applicability of different test systems to mutation analysis in patients with hematuria”, which contains only articles which clearly state that patients are afflicted by hematuria, and additionally briefly commented in the main text (lines 230-232): “Despite such constraints, many quantitative PCR-based diagnostic described below have been successfully applied to human biological fluids, e.g. whole blood [63], urine (see table 1) and urine samples of patients with hematuria (table S4)”.
Point 2: Conclusion: The authors mention that TERT-based assay is cost-effective. However, they did not describe cost of the assay. Please provide the basis of the conclusion.”
Response 2: We’ve provided these data in lines 416-423 in the updated version (last paragraph of the section 3.3. “Predictive significance of determining TERT promoter mutations in urine”): “The high recurrence rates of bladder cancer require frequent follow-ups involving expensive and invasive cystoscopic examination, thus further increasing the already high initial expenses for the management of bladder cancer [85,86]. The average costs of cystoscopy are around $206, and the cost of non-invasive urine cytology is around $56 [87]. In contrast, the cost of the NGS-based and ddPCR assays for detecting urinary TERT promoter mutations in bladder cancer developed by our group [10,11] is about 24€ per sample, and has therefore the potential to be easily implemented for cost-effective bladder cancer management strategies”.
References
- Mercier, B.; Gaucher, C.; Feugeas, O.; Mazurier, C. Direct PCR from whole blood, without DNA extraction. Nucleic acids research 1990, 18, 5908-5909.
- Srisuwan, W.; Tatu, T. A Simple Whole-Blood Polymerase Chain Reaction without DNA Extraction for Thalassemia Diagnosis. Hemoglobin 2018, 42, 178-183.

Reviewer 2 Report
The review on "Activating telomerase tert promoter mutations and their application for the detection of bladder cancer" is well written.
Few additional references should be included.
Page 3, lines 118-120: please add reference.
Page 3, lines 139-141: please update references and add "hTERT mRNA expression in urine as a useful diagnostic tool in bladder cancer. Comparison with cytology and NMP22 BladderCheck Test®. March-Villalba JA, et al. Actas Urol Esp. 2018 Oct;42(8):524-530".
Author Response
“The review on "Activating telomerase tert promoter mutations and their application for the detection of bladder cancer" is well written. Few additional references should be included”.
Point 1: Page 3, lines 118-120: please add reference.
Response 1: We have added next references «Preliminary findings indicate a high specificity in cellular models and in few healthy individuals and patients with malignancies other than bladder cancer [24-26], but this needs to be confirmed in large case-control studies»:
- Ludlow, A.T.; Robin, J.D.; Sayed, M.; Litterst, C.M.; Shelton, D.N.; Shay, J.W.; Wright, W.E. Quantitative telomerase enzyme activity determination using droplet digital PCR with single cell resolution. Nucleic acids research 2014, 42, e104;
- Vukašinović, A.R.; Kotur-Stevuljević, J.M.; Mlakar, V.; Sopić, M.D.; Cvetković, Z.P.; Petković, M.R.; Spasojević-Kalimanovska, V.V.; Bogavac-Stanojević, N.B.; Ostanek, B. Telomerase stability and evaluation of real-time telomeric repeat amplification protocol. Scandinavian journal of clinical and laboratory investigation 2019, 79, 188-193;
- Su, D.; Huang, X.; Dong, C.; Ren, J. Quantitative Determination of Telomerase Activity by Combining Fluorescence Correlation Spectroscopy with Telomerase Repeat Amplification Protocol. Analytical Chemistry 2018, 90, 1006-1013.
Point 2: Page 3, lines 139-141: please update references and add "hTERT mRNA expression in urine as a useful diagnostic tool in bladder cancer. Comparison with cytology and NMP22 BladderCheck Test®. March-Villalba JA, et al. Actas Urol Esp. 2018 Oct;42(8):524-530".
Response 2: This reference was added (reference number 32): «(1) the immunoassays based on detection of the Nuclear matrix protein 22 (NMP22® BC and its improved variant the NMP22® BladderChek®) [32]».
Point 3: Page 4, lines 158-160: please update references regarding use of urine biomarkers. Current references are from 2013, 2014, 2015.
Response 3: The reference (Shegay, P.V.; Zhavoronkov, A.A.; Gaifullin, N.M.; Vorob'ev, N.V.; Alekseev, B.Y.; Popov, S.V.; Garazha, A.V.; Buzdin, A.A.; Kaprin, A.D. Potentialities of MicroRNA Diagnosis in Patients with Bladder Cancer. Bull Exp Biol Med 2017, 164, 106-108.”) was added (reference number 42). Also, we provided more certain information about BC guidelines of American Urological Association and European Association of Urology (lines 157-163). “However, based on performance and cost considerations, none of the commercially available urine biomarkers to date are recommended as reliable diagnostic targets both by European Association of Urology (link to NMIBC guideline https://uroweb.org/guideline/non-muscle-invasive-bladder-cancer/#5; link to MIBC guideline https://uroweb.org/guideline/bladder-cancer-muscle-invasive-and-metastatic/#6) and American Urological Association (link to NMIBC guideline https://www.auanet.org/guidelines/bladder-cancer-non-muscle-invasive-guideline#x2517) for routine BC clinical management or for screening in high-risk populations [39-42]”.
Point 4: Page 10, Table 1. Please replace Urotherial with Urothelial.
Response 4: Replaced
Point 5: Page 12, Table 1. Please replace Uroterial with Urothelial.
Response 5: Replaced
Point 6: Please add reference: "Emerging immunotherapeutic strategies targeting telomerases in genitourinary tumors. Carrozza F, et al. Crit Rev Oncol Hematol. 2018 Nov;131:1-6".
Response 6: This reference was added (reference number 89): “With regards to potential therapeutic applications, it is worth noting that the region of the TERT promoter that frequently carries TERT promoter mutations in BC, which are absent in normal bladder cells, can presumably become the target of anti-cancer therapy including novel TERT-based immunotherapies, that could be tailored to patients whose tumors harbor these mutations [88,89]”.
